# Genetic Evidence for Endolysosomal Dysfunction in Parkinson’s Disease: A Critical Overview

**DOI:** 10.3390/ijms24076338

**Published:** 2023-03-28

**Authors:** Vidal Yahya, Alessio Di Fonzo, Edoardo Monfrini

**Affiliations:** 1Dino Ferrari Center, Department of Pathophysiology and Transplantation, University of Milan, 20122 Milan, Italy; vidal.yahya@unimi.it; 2Fondazione IRCCS Ca’ Granda Ospedale Maggiore Policlinico, Neurology Unit, 20122 Milan, Italy; alessio.difonzo@policlinico.mi.it

**Keywords:** Parkinson’s disease, genetics, lysosomes, endolysosomes, synaptic vesicles

## Abstract

Parkinson’s disease (PD) is the second most common neurodegenerative disorder in the aging population, and no disease-modifying therapy has been approved to date. The pathogenesis of PD has been related to many dysfunctional cellular mechanisms, however, most of its monogenic forms are caused by pathogenic variants in genes involved in endolysosomal function (*LRRK2*, *VPS35*, *VPS13C*, and *ATP13A2*) and synaptic vesicle trafficking (*SNCA*, *RAB39B*, *SYNJ1*, and *DNAJC6*). Moreover, an extensive search for PD risk variants revealed strong risk variants in several lysosomal genes (e.g., *GBA1*, *SMPD1*, *TMEM175*, and *SCARB2*) highlighting the key role of lysosomal dysfunction in PD pathogenesis. Furthermore, large genetic studies revealed that PD status is associated with the overall “lysosomal genetic burden”, namely the cumulative effect of strong and weak risk variants affecting lysosomal genes. In this context, understanding the complex mechanisms of impaired vesicular trafficking and dysfunctional endolysosomes in dopaminergic neurons of PD patients is a fundamental step to identifying precise therapeutic targets and developing effective drugs to modify the neurodegenerative process in PD.

## 1. Introduction

Parkinson’s disease (PD) is the second most common neurodegenerative disorder in the aging population [1,2,3,4]. It is clinically defined by the presence of bradykinesia in combination with either rest tremor and/or rigidity, and a clear beneficial response to dopaminergic therapy [5]. Neuropathologically, it is characterized by the loss of dopaminergic neurons in the substantia nigra (SN) and the presence of α-synuclein positive inclusions (Lewy bodies, LB) in surviving neurons [6,7,8]. At present, there are no approved treatments capable of slowing neurodegeneration in PD. Therefore, it is of paramount importance to shed light on the molecular mechanism causing PD neurodegeneration, because this knowledge is the indispensable prerequisite to identifying therapeutic compounds that can address the dysfunctional cellular machinery specific to this neurodegenerative disorder [9,10]. In the past two decades, PD etiopathogenesis has been linked with several deranged cellular mechanisms, ranging from mitochondrial impairment (*PRKN*, *PINK1*, *PARK7*) and ubiquitination defects (*FBXO7*) to dysfunction of the endolysosomal pathway (*LRRK2*, *VPS35*, *VPS13C*, *ATP13A2*) and synaptic vesicle trafficking (*SNCA*, *RAB39B*, *SYNJ1*, *DNAJC6*). In addition, significant parts of the risk genes associated with PD encode for endolysosomal and synaptic vesicle proteins, confirming a particular susceptibility of PD-related brain structures to the impairment of these pathways (Figure 1) [11,12,13,14,15,16,17,18].

Lysosomes are membrane-bound cytoplasmic organelles equipped with acid hydrolases whose major function consists in preserving cellular homeostasis by breaking down cellular (organized in autophagosomes) and extracellular (imported with endosomes) macromolecules and organelles into their fundamental components [19,20]. Lysosomal storage diseases (LSDs) are inborn metabolism defects often caused by loss-of-function (LOF) mutations in genes encoding lysosomal hydrolases. The consequent aberrant buildup of waste material causes lysosomal impairment, which can ultimately result in cellular dysfunction. Neurodegeneration is observed in association with several LSDs, clearly indicating that lysosomal impairment (or the toxic effect of mutant hydrolases) plays a relevant role in neuronal dysfunction and death [18,21]. Neurons display unique biological characteristics that may explain their increased susceptibility to the dysfunction of lysosomes and endosomal trafficking, such as their arborization and long projections, their post-mitotic state, and their highly complex interactions with other cells [22,23,24,25].

The scope of this review is to provide a critical overview of the genetic evidence supporting the central role of the endolysosomal pathway, that PD researchers can adopt as a solid starting point for future studies. We pursue this goal by reviewing monogenic PD forms and “strong” genetic PD risk variants associated with dysfunction of the endolysosomal pathway, then explore the rapidly evolving field of polygenic “weak” genetic risk variants affecting lysosomal genes, and finally present a unifying perspective linking this genetic evidence with the pathogenesis of PD [11,12,13,14,15,16,17].

## 2. Monogenic Causes of PD Associated with Endolysosomal and Vesicular Dysfunction

Early epidemiological studies revealed that 10–15% of PD patients have positive family history of the disease while most cases are sporadic [26]. Through family-based linkage analysis and, more recently, next-generation sequencing approaches, thirteen genes have been definitively implicated in the etiology of PD. Mutations in the *SNCA* [27,28], *LRRK2* [29,30,31], and *VPS35* [32,33] genes cause autosomal dominant forms, whereas mutations in the *PRKN* [34], *PARK7* [35], and *PINK1* [36] genes cause autosomal recessive forms. In addition, biallelic mutations in the *ATP13A2* [37], *PLA2G6* [38], *FBXO7* [39,40], *DNAJC6* [41], *SYNJ1* [42], and *VPS13C* [43] have been reported as rare causes of early-onset parkinsonism with atypical clinical features. Finally, *RAB39B* gene mutations have been associated with a form of X-linked levodopa-responsive parkinsonism in combination with various degrees of intellectual disability [44]. The possibility of relying on these established genetic PD forms represents a solid starting point to build hypotheses and generate genetic disease models for exploring the dysfunctional mechanisms underlying the PD pathology. In the following paragraphs, we focus on those Mendelian genes that support the role of impaired endolysosomal and vesicular function in the pathogenesis of PD (Figure 1).

### 2.1. LRRK2 (Leucine-Rich Repeat Kinase 2)

Mutations in the *LRRK2* gene represent the most common monogenic cause of PD, being accountable for around 3% of all cases [45]. The causality link between *LRRK2* mutations and autosomal dominant PD was unraveled in the early 2000s through linkage analysis and positional cloning [29,30,46,47] and was confirmed by many independent studies [31,48,49,50,51,52,53,54,55,56,57,58,59]. Although many candidate pathogenic variants were reported, pathogenicity was proven only for 8 of them, affecting the enzymatic domains of the Lrrk2 protein, including N1437H, R1441G/C/H/S, Y1699C, G2019S, and I2020T [60,61,62,63,64,65,66,67]. In addition, some other more common *LRRK2* variants were associated with increased risk of PD (e.g., A419V, R1628P, M1646T, and G2385R) [65,66,67].

The G2019S variant, the most common *LRRK2* pathogenic mutation in the western world, has a prevalence of 1% in all PD cases worldwide and is extremely frequent in Berber Arabs (up to 37% of PD patients) and Ashkenazi Jews (up to 23% of PD patients) [47,67,68,69,70,71]. The R1441C variant, the second most common *LRRK2* mutation, is more frequent in Italy [72]. The R1441G is a founder mutation in the Basque population (up to 46% in familial PD Basque patients) [73]. The R1628P and the G2385R risk variants are diffused in the Chinese population [74,75,76].

The clinical features of *LRRK2*-related PD are difficult to distinguish from those of idiopathic PD. From a motor perspective, it is characterized by bradykinesia, rigidity, and rest tremor, with unilateral tremor being the most common onset symptom [59,62,77]. While dystonia seems to be more frequent, cognitive decline, psychiatric symptoms, dysautonomia, and hyposmia are apparently under-represented compared with idiopathic forms [51,59,60,61,65]. Response to levodopa is optimal, though levodopa-induced dyskinesias (LID) are frequent [47,61]. Patients with *LRRK2*-related PD are probably more susceptible to inflammatory bowel diseases, strokes, and certain cancers (e.g., leukemia, myeloproliferative diseases, colon cancer, breast cancer) [65,78,79,80,81].

Neuropathologically, the majority of *LRRK2*-PD brains show α-synuclein-positive Lewy pathology [82], but some *LRRK2*-PD cases lack LB and others show different neuropathological features, such as tau pathology. Remarkably, a review of 55 cases of LRRK2-PD revealed that about half had tau pathology [83].

Lrrk2 is a large protein, with multiple domains and diverse functions in different cells and tissues. The two catalytic domains, GTPase and kinase, have been extensively investigated, with the caveat that most of the studies rely on overexpression of the protein. Lrrk2 was shown to phosphorylate the vesicular Rab GTPases Rab8A and Rab10 [14,84,85,86,87,88].

The role of *LRRK2* in PD has been thoroughly investigated in vitro (induced pluripotent stem cell, iPSC) and in vivo (*C. elegans*, Drosophila, and rodent models). Despite these models being able to identify several impaired mechanisms associated with Lrrk2 dysfunction, the precise link of mutated Lrrk2 to PD has not yet been fully elucidated [89,90,91,92]. Moreover, direct assessment of Lrrk2 endogenous subcellular localization is technically quite challenging, and the overexpression of the protein, fused with a tag in several studies, may be different from the physiological condition. Additionally, Lrrk2 subcellular localizations may change in different cell types and tissues, and under different conditions. Taking these premises into account, the localization of Lrrk2 in early and late endosomes has been repeatedly reported in several human cellular and animal models, supporting its involvement in endosomal trafficking and autophagy-lysosomal pathways [14,84,85,88,93]. In particular, Lrrk2 knockout murine models, which have no brain abnormalities, show peculiar abnormalities such as enlarged lamellar bodies (lysosome-related organelles) in lung cells and enlarged lysosomes with lipofuscin accumulation in kidneys, suggesting an important function of Lrrk2 in lysosomal homeostasis [94]. A direct interaction between Lrrk2 and α-synuclein has not been consistently demonstrated, and the murine models expressing mutant Lrrk2 do not recapitulate the most significant phenotypic features of the disease [95]. Pathogenic *LRRK2* mutations are deemed to be gain-of-function (GOF) variants that increase the kinase activity and consequently increase Rab8A and Rab10 phosphorylation, resulting in dysregulation of vesicular transport and mitophagy [14,66,85,88,96]. Interestingly, Rab29 protein (encoded by *RAB29*, a gene proposed as a risk locus for PD) was shown to play a role in the recruitment of Lrrk2 to stressed lysosomes [97,98]. Finally, recent evidence has revealed that mutated Lrrk2 alters ceramide metabolism and acts as a modifier of glucocerebrosidase level and enzymatic activity, further reinforcing its link with the endolysosomal pathway [99,100,101]. With all the caveats of the proposed models, the studies performed to date indicate a possible role of endolysosomal dysfunction in LRRK2-PD, which requires further validation in more appropriate models, such as iPSCs-derived neurons or organoids from patients carrying *LRRK2* mutations.

### 2.2. SNCA (Alpha-Synuclein)

*SNCA* was the first gene ever to be causally associated with PD. *SNCA* mutations were linked with PD in 1997, through linkage analysis in an Italian family (i.e., “Contursi kindred”). Shortly thereafter, α-synuclein was identified as a main component of LB [27,102,103,104]. To date, autosomal dominant PD was associated with *SNCA* duplications, triplications, and missense mutations including A30G/P, E46K, H50Q (debated role), G51D, and A53E/T/V [27,28,67,105,106,107,108,109,110,111,112,113,114,115].

Despite its great historical importance for understanding PD pathogenesis, *SNCA*-related parkinsonism is a very rare cause of disease, being accountable for up to 1–2% of familial and 0.2% of sporadic PD cases [67,112]. Clinically, it is characterized by earlier onset: around 43–48 years for A53T and duplications, around 31–36 years for triplications [67,112,116]. All classical motor features including bradykinesia, rigidity, tremor, postural instability, and gait disturbances are common. Dystonia has been reported to be more common in H50Q cases, myoclonus in A53E/T cases, pyramidal signs in G51D cases, and cerebellar symptoms in A30P cases. Additional atypical motor features include anterocollis, retrocollis, and alien limb syndrome. Among non-motor features, cognitive impairment is frequent for all mutations and psychosis is frequent for multiplications. Other overrepresented features are depression, anxiety, sleep disturbances, orthostatic hypotension, and urinary dysautonomia, while constipation and olfactory impairment are relatively rare [14,112,113]. Besides the typical SN degeneration, neuropathological studies showed several phenotypes including cortical Lewy pathology, diffuse Lewy body disease (LBD), α-synuclein glial cytoplasmic inclusions (GCIs), and CA2/3 hippocampus neuronal loss. Levodopa response is good, though motor fluctuation (MF) and LID are common [14,112,113].

The *SNCA* gene encodes α-synuclein, a protein with a poorly understood physiological function expressed in neurons and possibly oligodendrocytes, involved in the development and recycling of synaptic vesicles and plasticity of dopaminergic neurons [14,112,117,118,119,120,121]. *SNCA* mutations have been studied both with iPSC and in animal models, deepening the insight into its expression and aggregation but failing to reproduce neurodegeneration and motor symptoms in vivo [89,90].

Pathological α-synuclein tends to aggregate in variable size structures, from dimers to oligomers up to inclusion bodies, known as *Lewy bodies* in neurons, the pathological hallmark of PD [104,112,122,123,124]. Pathological α-synuclein aggregates are thought to cause neurodegeneration through endoplasmic reticulum (ER) and mitochondrial stress, protein misfolding and degradation, and synaptic and axonal dysfunction, impairing the retrograde transport of endosomes and autophagosomes to lysosomes located in the neuronal body [123,125,126,127,128,129,130,131]. Multiplications of *SNCA* increase the expression of α-synuclein and consequently its tendency to form pathological aggregates, while, curiously, all identified missense variants affect the N-terminal region of α-synuclein, possibly impacting its ability to bind at cellular membranes [112,132,133]. It is relevant to note that α-synuclein in physiological conditions is mainly degraded by the autophagosome–lysosome pathway. Therefore, a disruption of this pathway may cause α-synuclein aggregation and accumulation in neurons [13]. Supporting this hypothesis, PD-causing mutations such as A30P and A53T are probably associated with an impairment of α-synuclein degradation through autophagy [134,135] (Figure 1).

### 2.3. VPS35 (Vacuolar Protein Sorting 35 Ortholog)

In 2011, two independent studies using exome sequencing and linkage analysis revealed the link between the D620N mutation of *VPS35* and autosomal dominant PD [32,33]. Since then, many variants have been associated with PD (R32S, G51S, M57I, I241M, P316S, R524W, I560T, M57I, H599R, M607V, L774M); however, their pathogenicity is still unconfirmed [136,137,138].

The clinical features of *VPS35*-related PD are similar to those of typical PD, with a relatively younger age of onset (46.9 ± 8.6 years). The reported motor features are usually classical (bradykinesia, rigidity, rest tremor) with asymmetric onset. Regarding non-motor features, dysautonomia, psychosis, and hallucinations are rare, and cognitive decline is observed in 15–30% of cases. The therapeutic response to dopaminergic therapy is good, though wearing-off and levodopa-induced dyskinesias appear in around 80% of cases [138,139,140,141]. Neuropathological findings remain to be determined [138,139,140,141].

The *VPS35* gene encodes a subunit of the heteropentameric retromer complex, located at the endosomal membrane, where it facilitates the endosome-to-Golgi and the endosome-to-plasma membrane transport [138,142,143]. Rodent- and iPSC-based studies are providing greater insight into *VPS35*-related neurodegeneration [90,92,138,144,145]. The D620N variant was found to be associated with impaired autophagy, possibly due to abnormal sorting of the ATG9A autophagy receptor and decreased autophagosome formation [138,144]. Furthermore, it determines a defective sorting of cathepsin D (encoded by *CTSD*, see below), resulting in lysosomal dysfunction, which potentially implies an impaired degradation of aggregation-prone proteins such as α-synuclein [138,145]. A third mechanism may be identified in mitochondrial degradation, since Vps35 physiologically facilitates mitophagy via mitochondrial-derived vesicles and the mutated Vps35 might enhance the activity of Dlp1, a protein involved in mitochondrial fission, thus resulting in mitochondrial fragmentation [138,146,147]. Vps35 D620N mutation was shown to enhance Lrrk2-mediated phosphorylation of Rab10 as well as autophosphorylation, suggesting that Vps35 may be an upstream regulator of Lrrk2. Overexpression of wildtype Vps35 was demonstrated in flies and murine models to rescue retromer-mediated defects, such as lysosomal enlargement, caused by Lrrk2 G2019S overexpression or Rab29 knockdown. These observations support a possible shared mechanism of Vps35, Lrrk2, and Rab29 in PD pathogenesis [138,148]. Importantly, the demonstration of a common shared altered mechanism of these proteins involving vesicular and endolysosomal trafficking will require further validation in human-based neuronal models hopefully capable of recapitulating PD pathology. In this direction, recent studies of human iPSC-derived neurons carrying the *VPS35* D620N mutation showed decreased autophagic flux [149].

### 2.4. VPS13C (Vacuolar Protein Sorting 13 Homolog C)

Biallelic *VPS13C* mutations cause autosomal recessive early-onset PD (EOPD) [43,150]. The link between *VPS13C* mutations and PD was identified through a genome-wide association study (GWAS) in 2014 and confirmed in 2016 through homozygosity mapping and exome sequencing study of consanguineous families, when the homozygous c.8445 + 2T > G variant was identified in a Turkish patient with PD [43,151].

To our knowledge, only 18 *VPS13C*-related PD cases have been clinically and genetically described to date, with earlier onset (37.5 ± 10.5 years of age) and faster clinical progression compared to idiopathic PD [152]. From a motor perspective, *VPS13C*-related PD is generally characterized by classical features such as bradykinesia, rigidity, rest tremor, freezing, and postural instability, as well as by distinct features including dystonia and, more rarely, pyramidal signs. Concerning non-motor features, dysautonomia, hyposmia, early cognitive decline, and visual hallucinations are presented. Dopaminergic therapy is usually effective, though MF and LID are common [43,150,152,153,154,155,156,157]. Neuropathological studies have shown diffuse LBD [43,150,152,153,154,155,156,157].

The gene *VPS13C* encodes an intermembrane lipid transfer protein localized at contact sites between the ER and late endosomes/lysosomes and on the outer mitochondrial membrane [43,158]. Vps13C regulates lysosomal homeostasis and controls mitophagy, modulating the Pink1/Parkin pathway in cellular models [43,159]. The neurodegeneration associated with the loss of *VPS13C* function thus seems primarily attributable to an alteration of lysosomal homeostasis and an upregulation of Pink1/Parkin-dependent mitophagy [43,153,154,159]. However, whether this impaired mechanism observed in vitro reflects the pathological process in *VPS13C*-PD brains is far from being clear yet.

### 2.5. ATP13A2 (ATPase Cation Transporting 13A2)

Biallelic *ATP13A2* mutations have been associated with a multitude of phenotypes including Kufor–Rakeb disease (KRD), neuronal ceroid lipofuscinosis, hereditary spastic paraplegia, and an amyotrophic lateral sclerosis-like form [37,160,161]. KRD is a rare AR, levodopa-responsive, rigid-akinetic parkinsonism with atypical features including pyramidal signs, supranuclear gaze palsy, and cognitive decline. It was first described in 1994 in a Jordanian family, and was associated with *ATP13A2* through linkage analysis in 2006 [37,162,163].

Around 50 cases of KRD have been reported to date. The age of onset is usually in the second decade of life. Additional clinical features may include tremor, facial-faucial-finger mini myoclonus, oculogyric dystonic spasms, slow saccadic eye movement, peripheral neuropathy, and visual hallucinations [163,164,165,166,167,168,169,170,171,172,173]. Brain MRI may display generalized brain atrophy with bilateral putaminal and caudate iron accumulation [165,174,175,176]. After an initial response, MF and LID rapidly occur [163,164,177,178]. While the KRD phenotype is generally associated with truncating *ATP13A2* mutations, missense mutations in *ATP13A2* tend to be associated with a clinical form similar to EOPD [177,179,180,181,182]. A single neuropathologic case of KRD was reported, showing widespread neuronal and glial lipofuscin accumulation with no LB-type inclusions and absence of α-synuclein-positive pathology [183].

*ATP13A2* encodes a lysosomal P-type ATPase which has been associated with the homeostasis of metal cations (e.g., Fe^3+^, Mn^2+^, Zn^2+^), mitochondrial clearance, and, possibly, α-synuclein detoxification [171,184,185]. In cell lines, the endogenous level of Atp13a2 repeatedly appeared below immunodetection, despite using different currently available Atp13a2 antibodies. Therefore, most of the studies showing co-localization with lysosomal and late endosomal/intraluminal vesicle markers are based on overexpression of the protein. Atp13a2-deficient mice show sensorimotor deficits, and accumulation of insoluble α-synuclein in the brain, which is exacerbated by overexpression of the human wildtype α-synuclein [186]. Apoptosis dysregulation, mitochondrial dysfunction, and ER stress have emerged from animal and cellular models. Moreover, Atp13a2 LOF determines lysosomal dysfunction with defective polyamine export and autophagosome dysfunction, as effectively explored both in vivo and in vitro [176,187,188,189,190,191].

### 2.6. RAB39B (RAB39B, Member RAS Oncogene Family)

LOF mutations in *RAB39B* were identified as a cause of X-linked recessive (XLR) early-onset parkinsonism (EOP) in 2014 through linkage analysis in three Australian brothers presenting intellectual disability (ID) with EOP, and confirmed via linkage analysis and direct *RAB39B* sequencing in a Wisconsin family with 13 males presenting a similar phenotype [44].

The clinical features of *RAB39B*-related disease range from classical PD to syndromic forms with various combinations of EOP (akinetic-rigid or with tremor), dystonia, ID (delayed speech, learning difficulties), autistic spectrum disorders, seizures, frontal lobe reflexes, macrocephaly, strabismus, and short stature. Brain magnetic resonance imaging (MRI) may display SN and globus pallidus iron deposition, while brain computerized tomography (CT) may show symmetrical globus pallidus calcification. Response to levodopa is often present, though saddled with MF and LID [44,192,193,194,195,196,197,198,199,200,201,202,203]. Neuropathological findings include SN and cortical neuronal death and LBD, SN tau-positive neurofibrillary tangles, and basal ganglia axonal spheroids [44,195,196,198,202,203,204]. It is to be noted that *RAB39B*-related parkinsonism is caused by large-scale deletions, splicing abnormalities, and frameshift, nonsense, and missense mutations resulting in LOF. Duplications and triplications of *RAB39B* may result in a pathologic GOF leading to complex syndromes with ID and behavioral abnormalities [200,203,204,205,206,207,208].

*RAB39B* encodes a neuronal GTPase involved in vesicular trafficking and recycling between synaptic terminals, endosomes, and the Golgi apparatus. Rab39B is thought to control many cellular functions including α-synuclein homeostasis and GluA2/GluA3 AMPAR subunit trafficking from the ER to the Golgi apparatus, as suggested by several murine, iPSC, and isogenic human embryonic stem cell models [44,193,194,195,200,202,209,210].

### 2.7. SYNJ1 (Synaptojanin 1)

Biallelic *SYNJ1* mutations were linked with autosomal recessive EOPD in 2013 through homozygosity mapping and exome sequencing, in two parallel studies describing an Iranian and an Italian family [42,211].

To date, 34 patients originating from 20 families with biallelic *SYNJ1* missense, nonsense, frameshift, or splicing mutations have been reported to constitute a heterogeneous multitude of phenotypes ranging from EOPD or atypical parkinsonism (13 families) up to severe epileptic encephalopathies without parkinsonism (seven families) [42,211,212,213,214,215,216,217,218,219,220,221,222,223,224].

*SYNJ1*-related parkinsonism is generally characterized by disease onset in the third decade of life. Classical features include bradykinesia, rest tremor, and postural instability. A variety of atypical features have been reported including vertical gaze palsy, eyelid apraxia, diplopia, dysarthria/anarthria, hypophonia, dysphagia, drooling, oromandibular tremor, dystonia, pyramidal signs, cognitive decline, and seizures. In some cases, neuroimaging was reportedly normal, while in other cases it displayed variable abnormalities including cortical atrophy/hypometabolism, caudate hypometabolism, and bilateral nigrostriatal dopaminergic denervation. Levodopa response was variable and a multitude of adverse effects including dyskinesia, dystonia, and postural hypotension have been reported. Among them, five families with various ethnicities carried the R258Q missense variant, which is therefore thought to be a possible mutational hotspot [42,211,212,213,214,215,216,217,218,219,220,221,222,223,224]. A single autoptic examination displayed neuronal loss in SN without LBD, neurofibrillary degeneration, and tau protein staining in cell bodies and axonal hillocks [212].

*SYNJ1* encodes an inositol phosphatase that allows the shedding of clathrin coats and other endocytic factors from their membranes in the Golgi apparatus, endosomes, and plasma membrane, playing a fundamental role in synaptic vesicle endocytosis and autophagy [42,225,226,227,228,229]. The pathogenesis of *SYNJ1*-related parkinsonism has been studied in zebrafish, Drosophila, and rodent models [92,228,230,231,232]. Loss of *SYNJ1* function causes synaptic autophagy and transmission defects manifesting with delayed synaptic vesicle endocytic recycling and accumulation of clathrin-coated vesicles [228,229,231,232].

### 2.8. DNAJC6 (DnaJ Heat Shock Protein Family Member C6)

In 2012, biallelic *DNAJC6* mutations were linked with juvenile-onset autosomal recessive parkinsonism through homozygosity mapping and exome sequencing in a Palestinian family [41].

To date, 21 cases of *DNAJC6*-related parkinsonism have been reported, carrying biallelic missense, nonsense, frameshift, or splicing mutations [41,233,234,235,236,237,238,239]. *DNAJC6*-related parkinsonism is typically characterized by disease onset in the second decade of life and rapid clinical progression until loss of walking within around 10 years from onset. Initial symptoms tend to be rest tremor and bradykinesia, followed by rigidity and postural instability. Although classic PD is a possible form of *DNAJC6*-related parkinsonism, there usually are atypical manifestations such as dystonia, pyramidal signs, postural tremor, dysarthria, anarthria, epilepsy, ID, and psychosis. Levodopa response is extremely variable and often burdened by dyskinesias. No consistent neuropathological features have yet been described [67,237,240].

*DNAJC6* encodes a neuron-specific isoform of auxilin-1, a co-chaperone involved in clathrin-coat detachment after endocytosis, to ease vesicle recycling, the role of which was explored using human embryonic stem cells with CRISPR-Cas9–mediated gene editing [14,67,241,242,243]. *DNAJC6* LOF disrupts synaptic vesicle endocytosis and induces α-synuclein overexpression, thus possibly leading to dopaminergic neurodegeneration [241,242].

In conclusion, monogenic evidence seems to indicate two different groups of genes within the endolysosomal pathway: those more involved in lysosomal-autophagic functions and those encoding for proteins related to the machinery of synaptic vesicles, which are considered a specialized form of recycling endosome present in neurons. The clinical phenotype and neuropathologic findings of the first group seem to recapitulate better the idiopathic form of PD (e.g., *LRRK2*, *VPS35*, *VPS13C*). Conversely, the second group is characterized by atypical phenotypes and, as far as is known, neuropathology (e.g., *SYNJ1* and *DNAJC6*). Therefore, one could speculate that the best models to identify disease mechanisms and therapeutic compounds for the common idiopathic form of PD should be based on the first group of genes, to which *GBA1* should also be added (see below). In this context, *SNCA* and *RAB39B* probably deserve a separate discussion: although they seem functionally more involved in synaptic function, they display widespread LBD, more in line with idiopathic forms of PD. Therefore, these genes may represent a bridging mechanism between the two groups described above (Figure 1 and Figure 2, Table 1).

## 3. Genetic Risk Factors for PD Associated with Endolysosomal Dysfunction

Despite these great advances in the field of PD Mendelian genetics, a monogenic cause can be identified in only a minor fraction of PD cases. In general, non-monogenic PD is thought to be a multifactorial disorder influenced by genetic and environmental factors. The emergence of new technological approaches and the increasing sizes of international PD cohorts are leading to the identification of common variants with small effects contributing to PD [244,245]. Pathway analysis of the genes in which these variants are located has shown the involvement of several pathways including mitochondrial biology, inflammation/immune response, and lysosomal-autophagic functions [246,247]. Here, we focus on the risk genes involved in the endolysosomal function (Figure 1).

### 3.1. GBA1 (Glucosylceramidase Beta 1 or Glucocerebrosidase)

Pathogenic variants in the *GBA1* gene represent the most common genetic risk factor for PD: it is estimated that about 8.5% of PD patients worldwide carry a *GBA1* mutation. *GBA1* carriers display a five- to seven-fold increased risk of developing PD, with a lifelong penetrance of 10–30% [45,151,248,249,250,251,252,253,254,255]. This is especially relevant for specific groups with a high frequency of *GBA1* mutations, such as Jews (Ashkenazi Jews PD: 10–31%, non-Ashkenazi Jews PD: 2.9–12%), whereas the frequency in the general population is around 1% [248,256].

The *GBA1* gene encodes the glucosylceramidase beta 1 (GCase), a lysosomal hydrolase that breaks glucosylceramide into glucose and acylsphingosine [257,258]. Biallelic *GBA1* mutations cause Gaucher’s disease (GD), a lysosomal storage disorder biologically characterized by a significant reduction in GCase activity, which leads to toxic accumulation of glucosylceramide and glucosylsphingosine [259]. Monoallelic *GBA1* mutation carriers do not clinically develop GD nor present a pathological accumulation of glucosylceramide and glucosylsphingosine [250,260]. The link between monoallelic *GBA1* mutations and PD emerged between 1985 and 1988 when parkinsonism was described as a possible neurological manifestation in GD patients and their relatives carrying heterozygous *GBA1* mutations [260,261,262,263,264]. This link was confirmed by subsequent large-scale genetic studies, including GWAS [11].

The clinical features of *GBA1*-related PD are similar to those of idiopathic PD [250,256,265,266,267,268,269]. However, the average age of onset tends to be slightly earlier (1 to 6 years), clinical progression is generally faster, and survival is shorter [250,270,271,272,273,274,275,276]. Non-motor symptoms including hyposmia, constipation, urinary dysautonomia, orthostatic hypotension, sleep disturbances, depression, anxiety, and especially cognitive impairment (with a decline in working memory, visuospatial, and executive function), are more frequent and impactful [250,256,268,269,271,273,274,275,276,277,278,279,280,281,282,283]. Neuropathological features of *GBA1*-related PD precisely resemble those of idiopathic PD, with dopaminergic neuron loss and LBD in SN. Cortical LBD was also reported [256,264,269,271,284,285,286,287]. Levodopa response is effective, although MF and LID occur earlier and more frequently [269,276,288] (Figure 2).

Point mutations and complex alleles in *GBA1* (resulting from rearrangement with the pseudogene *GBAP1*) are both reported as strong genetic risk factors for PD. Missense mutations are most represented, including the main causative mutations for GD (e.g., N370S and L444P) as well as PD risk variants that are not associated with GD (e.g., E326K and T369M) [256,289,290,291,292,293,294]. Severity classification for *GBA1* mutations considers their impact on the GD phenotype and correlates with the residual GCase activity. The degree of severity of *GBA1* mutations, which can be defined as “severe” or “mild”, has a differential effect on penetrance, age at onset, and clinical progression of PD [282,295]. However, it is still not completely clear whether this severity classification derived from GD patients can be automatically translated to the PD field [256,267,271,282,295,296,297].

The disease mechanisms of *GBA1*-related PD are yet to be understood and may not perfectly match with those of GD, as there are PD risk variants that do not cause GD in homozygosis (e.g., E326K and T369M), and since no definitive correlation has been demonstrated to date between GCase activity and *GBA1*-related PD [256,290,292,293,298,299]. Nevertheless, the GCase protein was identified in 32–90% of Lewy bodies in patients with *GBA1*-related PD or Dementia with Lewy bodies (DLB) and less than 10% of Lewy bodies in patients with idiopathic PD, suggesting a possible interaction between GCase and α-synuclein [12,17,256,269,300]. Several pathogenetic hypotheses for *GBA1*-related PD have been proposed to date, following large studies on PD brains, Drosophila, murine models, cell models including midbrain-like organoids, and iPSC-derived dopamine neurons [256,301] The suggestions include different mechanisms that may depend on both gain- and loss-of-function mutations, including ER stress, ER-Golgi transport impairment, α-synuclein clearance impairment, lysosomal and autophagic dysfunction, lipid homeostasis disruption, mitochondrial dysfunction, and neuroinflammation [256,269,301,302,303,304,305,306,307,308,309,310,311,312]. An intriguing hypothesis is that inactive GCase located on the surface of the lysosomal membrane may change the composition of glycolipids of the membrane affecting the lysosomal internalization of α-synuclein for degradation [97,313]. Another important study showed that lysosomal glucosylceramide accumulation due to GCase deficiency directly interacts with α-synuclein and leads to its accumulation, and vice versa, so that the pathological buildup of α-synuclein may inhibit the transport of GCase to the lysosome, inducing a toxic vicious cycle for neurons [303].

Pathogenic *GBA1* variants alone are not sufficient to develop PD, because in groups of patients carrying the same mutations, some develop PD while others do not. Among *GBA1* mutation carriers, the development of PD probably depends on other genetic and environmental factors. In this context, two risk-modifier variants (rs356219 in *SNCA* and rs1293298 in *CTSB*) were initially identified and relatively little impact was attributed to them, while a larger cumulative impact was reported for common variants affecting genes involved in lysosomal function [12]. Furthermore, other *SNCA* variants and *TMEM175* M393T (see below) may influence the onset age of *GBA1*-PD [12]. Extending the research into risk-modifying factors to rare predicted deleterious variants in lysosomal genes, the largest contribution to the development of PD in *GBA1* mutation carriers was attributed to a second deleterious variant in *GBA1* or a deleterious variant in genes associated with mucopolysaccharidoses, evidencing the importance of the overall lysosomal burden in the development of PD [12,17,314].

### 3.2. SMPD1 (Sphingomyelin Phosphodiesterase 1 or Acid-Sphingomyelinase)

The gene *SMPD1* encodes acid sphingomyelinase (ASMase), a lysosomal hydrolase that breaks sphingomyelin into ceramide and phosphocholine [280,281,282,283]. Biallelic *SMPD1* mutations cause Niemann–Pick disease (NPD), an LSD characterized by sphingomyelin accumulation, with a heterogeneous range of phenotypes including a severe infantile multisystemic disorder with neurodegeneration (type A) and a later-onset form with absent or minimal neurologic involvement (type B) [315,316,317].

*SMPD1* variants L302P and P330fs, highly prevalent among Ashkenazi Jews, were repeatedly associated with PD in this population through case–control studies, as about 1.5% of PD patients carried these mutations, compared with 0.4% of controls [318,319,320,321,322,323]. Previous studies showed that reduced ASMase levels lead to α-synuclein accumulation [323]. It is therefore possible to assume that these mutations increase the risk of developing PD by reducing lysosomal localization of ASMase and thus causing accumulation of pathological α-synuclein [323].

Moreover, several other studies on European and Chinese populations identified additional *SMPD1* variants suspected to be associated with PD [254,324,325,326]. Nevertheless, it is important to note that only some of the *SMPD1* mutations causing NPD type A/B were demonstrated to be associated with PD [323].

### 3.3. TMEM175 (Transmembrane Protein 175)

GWAS approaches in PD cases repeatedly identified an association peak on chromosome 4 (TMEM175/GAK/DGKQ locus) representing the fourth strongest risk locus. Two coding variants in this locus localized in the gene *TMEM175* (M393T and Q65P) were found to be associated with PD. Transmembrane protein 175 is an integral membrane protein involved in potassium ion transmembrane transport in endolysosomes. Interestingly, the M393T variant was shown to be associated with reduced GCase activity. Therefore, *TMEM175* variants are probably responsible for the GWAS association in the TMEM175/GAK/DGKQ locus, as also supported by a Mendelian randomization study [12,13,155,327,328,329,330,331,332,333].

### 3.4. SCARB2 (Scavenger Receptor Class B Member 2)

*SCARB2* variants have been repeatedly identified as risk factors for PD [12,334,335]. *SCARB2* encodes the scavenger receptor class B member 2, which, among other endolysosomal activities, transports the GCase protein from the Golgi apparatus to the lysosome. However, unlike the *TMEM175* M393T variant, the identified *SCARB2* risk variants do not seem to affect GCase activity [335,336,337,338].

### 3.5. Polygenic “Lysosomal Burden”

An important genetic study using a sequence kernel association test (SKAT-O) investigated deleterious variant burden among lysosomal storage disorder genes using whole exome sequencing data from vast cohorts of PD cases and control subjects. The authors identified a significant burden of rare, potentially damaging lysosomal gene variants in PD patients compared with controls. The association persisted even when the *GBA1* gene was excluded from the analysis, suggesting a significant “lysosomal burden” in idiopathic forms of PD. Consistent results were obtained in two independent replication cohorts. The lysosomal genes found to drive the association were *GBA1*, *SMPD1*, *CTSD*, *SLC17A5*, and *ASAH1*. Remarkably, in the discovery cohort, the majority of PD cases had at least one deleterious variant in an LSD gene, and 21% carried multiple damaging alleles [254]. Additional clues linking lysosomal dysfunction and “idiopathic” PD include the identification of a deficiency in the activity of the GCase enzyme in blood, CSF, and brain structures not only in *GBA1*-PD but also in idiopathic forms [339,340].

A recent study investigated the role of deleterious variants affecting LSD genes in modulating the penetrance of *GBA1* risk variants in PD. The analysis in the discovery cohort revealed a significantly increased burden of deleterious variants in *GBA1*-PD patients compared to healthy *GBA1* mutation carriers. Moreover, the authors demonstrated that the two strongest modifiers of *GBA1* penetrance were a second variation in *GBA1* (5.6% vs. 1.4%) and variants in genes causing mucopolysaccharidoses (6.9% vs. 1%) [17]. Furthermore, the largest GWAS on PD to date, based on about 40,000 PD cases including genetic and idiopathic forms, with 1.4M controls, found that an SNP in the gene *GALC* (rs979812) is associated with PD [11]. *GALC* encodes galactosylceramidase, a lysosomal hydrolase involved in ceramide catabolism, similar to *ASAH1*, *GBA1*, and *SMPD1* [13,21,341,342]. Interestingly, the *GALC* rs979812 variant seems to be associated with increased enzymatic activity of galactosylceramidase [343].

Enrichment analysis based on the same GWAS study revealed that pathways related to lysosomal function are among the most significantly enriched [11]. Other endolysosomal genes identified through a GWAS approach were *BAG3*, *CTSB*, *GPNMB*, *GRN*, *GUSB*, *HIP1R*, *LAMP3*, *NEU1*, *RAB29*, *SCARB2*, *SH3GL2*, and *TMEM175* [13]. Another GWAS demonstrated significant effects of *BAG3*, *GBA*, *LAMP3*, *SCARB2*, *SNCA*, and *TMEM175* loci on age at onset of PD [12]. However, it is worth noting that large GWAS, for which perfect matching of cases and controls is virtually impossible, can result in spurious associations because of population stratification. This is also a major concern when studying recently admixed populations, particularly regarding variants with very small effect size [344]. Moreover, the scope of GWAS is limited to identifying genetic associations between PD status and genetic variants, tagging genomic regions encompassing several candidate genes. The “true” causal genes in each identified locus and the mechanisms by which they confer increased risk of PD often remain unclear. This important issue can be addressed by complementing GWAS data with quantitative trait loci (QTL) datasets correlating risk genotypes with gene expression, methylation, and proteomic data (i.e., Mendelian randomization studies—MRS); this knowledge of the molecular mechanisms by which genetic variants localized in PD risk loci increase the disease risk is a key step to translating genetic evidence into possible therapeutic targets. Remarkably, the link between endolysosomal impairment and synaptic dysfunction within PD derives also from these approaches, which have recently confirmed the causal role of several “lysosomal” and “synaptic” genetic hits (i.e., *ARSA*, *CTSB*, *GALC*, *IDUA*, *RAB29*, *RAB7L1*, *SH3GL2*, *SMPD1*, *STX1B*, *TMEM175*, *VAMP42*, and *ZSWIM7*) [333,343,345,346,347,348,349].

## 4. Unifying Perspective and Conclusions

The previous paragraphs support the hypothesis that genetic variants of genes involved in endolysosomal function are major determinants of PD pathogenesis. Nonetheless, the existence of PD genes and risk variants in genes not directly involved in this pathway suggests that isolated dysfunction of the endolysosomal pathway is unable to explain the full picture, and additional genetic or environmental hits are important factors at play, especially in late-onset non-monogenic forms (Figure 1).

It is an established fact that neuropathological studies of PD brains display abnormal aggregates of misfolded proteins, among which the major actor seems to be aggregated α-synuclein. Other pathological proteins have also been described, for example, aggregated tau in Lrrk2-PD patients [83] (Figure 1).

A dysfunctional endolysosomal system may determine the reduction of α-synuclein turnover, increasing its cytoplasmic concentration and ultimately promoting the formation of oligomeric and fibrillary species, which are considered damaging for dopaminergic neurons [350,351]. An additional link to endolysosomal dysfunction is the presence of membranaceous structures in Lewy bodies, the pathological hallmark of the disease [352]. Moreover, dysfunctional endomembrane trafficking may induce the exploitation of alternative α-synuclein clearance routes, involving the plasma membrane and exocytosis, and ultimately promoting the spread of disease (“prion-like spread hypothesis” of PD) [353,354]. In our view, a vicious circle started by abnormal protein accumulation and subsequent activation of dysfunctional endolysosomal and autophagic processes may be the culprit of this neurodegenerative disease (Figure 1). However, it should be noted that it remains open to debate whether α-synuclein aggregates should be portrayed as the cause or a result of the neurodegenerative process in PD. Taking this forward, a deficiency of functional monomeric α-synuclein has been proposed as a pathogenetic mechanism underlying PD (“proteinopenia” vs. “proteinopathy” hypothesis) [355]. In any case, the strong genetic evidence supporting the role of vesicular and endolysosomal impairment in PD should be carefully considered when formulating any new hypothesis for PD pathogenesis.

The causes of the selective vulnerability of dopaminergic neurons in PD are still unclear. Some peculiar characteristics of these neurons, such as the high metabolic-energetic demands, pace-making activity for the regulated release of dopamine, the pro-oxidant metabolism of this neurotransmitter, and the presence of long and complex arborizations, may suggest possible mechanisms [356]. In this scenario, normal endolysosomal and synaptic functions seem particularly important in the continuous process of recycling synaptic vesicles and the maintenance of long and complex neuronal projections, which are known to recede in PD brains in a dying-back process [357,358,359]. Conversely, high energy needs and oxidative stress management may indicate the mitochondrion as the prime suspect. Unsurprisingly, genetic evidence supports a very significant role also of the mitochondrial pathway (e.g., *PRKN* and *PINK1* mutations) [356].

This review highlights the great importance of research in the field of PD genetics. Starting from the genetic findings accumulated over the past few decades and solidly based on that knowledge, a great number of in vitro and in vivo functional studies of disease models have been conducted, leading to an increased understanding of PD disease mechanisms. However, it is likely that some of the abnormalities observed in these models, although reproducible, are not pathogenetically linked to PD in humans. Some of the animal models do not present the neuropathological or phenotypical features of PD, calling into question the significance of the findings. It is probable that more advanced models, such as patient-derived midbrain organoids carrying specific gene mutations, represent more suitable tools to recapitulate the disease process that occurs in PD patients.

In conclusion, multiple lines of genetic evidence support a major role of vesicular and endolysosomal dysfunction in the pathogenesis of PD. As we have shown, several genes involved in these pathways have been demonstrated to be causative in monogenic forms of PD or have been shown to be associated with increased risk of PD. The crucial role of the endolysosomal system in PD has also generated therapeutic prospects for PD treatment. Indeed, several studies have suggested that the maintenance or upregulation of lysosomal activity may protect against neurodegeneration in PD [360]. In this view, promising therapeutic endolysosomal and autophagic targets include enzymatic activity enhancement (e.g., ambroxol), substrate reduction agents (e.g., venglustat), lysosomal activation (e.g., farnesyltransferase and mTOR inhibitors), and autophagic induction (e.g., trealose and nilotinib) [360]. Currently, many active therapeutic trials in humans have targetted this pathway in patients with genetic and idiopathic forms of PD [269,361,362,363,364,365]. Hopefully, an increased understanding of the specific mechanisms by which dysfunction of these pathways causes PD may allow for the development of targeted drugs that can modify the invariably progressive course of PD.

## Figures and Tables

**Figure 1 ijms-24-06338-f001:**
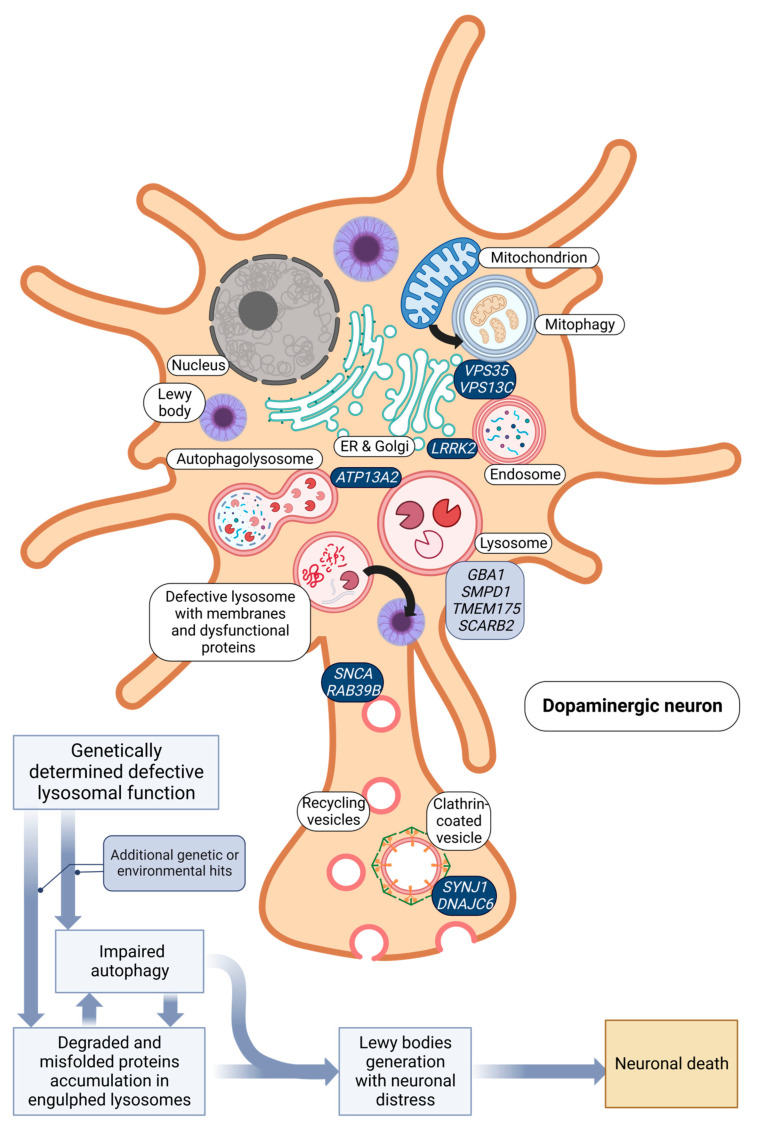
Graphic representation of a dopaminergic neuron illustrating the relevant intracellular organelles, the involved monogenic PD genes (dark blue), and PD risk genes (light blue). In the represented model, a genetically determined impairment of the endolysosomal function in association with additional genetic and environmental pathogenic hits causes a defect in the degradation of misfolded proteins (including α-synuclein), which induces a compensatory autophagic response to address the disposal of the pathological proteins. However, a defective endolysosomal pathway implies an ineffective autophagic response, thus triggering a self-feeding vicious cycle of abnormal protein accumulation by dysfunctional engulfed lysosomes. As a consequence, the accumulation of misfolded aggregated proteins and engulfed lysosomes–autophagosomes leads to the progressive formation of pathological inclusions (Lewy bodies) and ultimately to neuronal distress and death. Created with BioRender.com.

**Figure 2 ijms-24-06338-f002:**
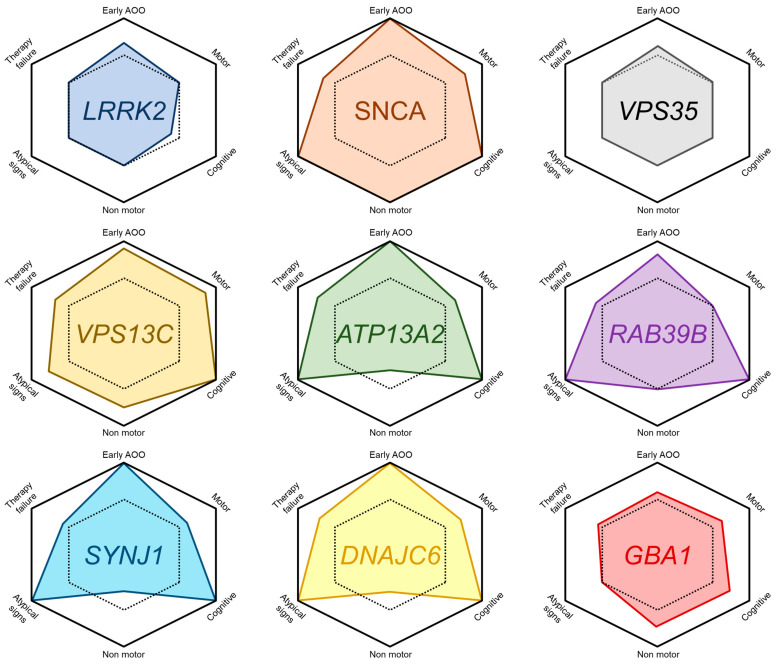
Phenotypic comparison between each monogenic form and idiopathic PD (represented by the dashed hexagon). The considered features are: age of onset (AOO), motor signs (bradykinesia, rigidity, rest tremor), cognitive problems (including neuropsychiatric symptoms and intellectual disability in the case of *RAB39B*), non-motor symptoms of PD including (dysautonomia, hyposmia, REM sleep behavior disorder), atypical signs (e.g., epilepsy, spasticity, dystonia, dysmorphisms), and therapy failure (poor response to levodopa treatment or early onset of LID and MF).

**Table 1 ijms-24-06338-t001:** Monogenic causes of PD (including *GBA1*-PD).

Gene (Inheritance)	Biology	Phenotype	Levodopa Response	Neuropathology
*LRRK2* (AD)	Missense GOF disrupts vesicular transport and mitophagy	Classical PD, cancer	Good, with frequent LID	LBD of SN, occasional Tau pathology
*SNCA* (AD)	Multiplications increase α-synuclein expression, missense impair degradation	Earlier onset, dystonia, pyramidal signs, cerebellar signs, cognitive impairment, psychosis, depression, anxiety, orthostatic hypotension, urinary dysautonomia	Good, with frequent MF and LID	SN and cortical LBD, α-synuclein GCIs
*VPS35* (AD)	D620N impairs autophagosome formation (ATG9A), lysosomal function (cathepsin D), and mitophagy; increases Lrrk2 substrates phosphorylation	Classical PD with earlier onset; cognitive and dysautonomic features are rare	Good, with frequent MF and LID	NA
*VPS13C* (AR)	LOF alters lysosomal homeostasis and promotes Pink1/Parkin mitophagy	Parkinsonism with dystonia, rare pyramidal signs, early cognitive decline, visual hallucinations	Good, with frequent MF and LID	Diffuse LBD
*ATP13A2* (AR)	LOF causes ER stress, lysosomal dysfunction (defective polyamine export), and autophagosome dysfunction	From EOPD (usually missense variants) to KRD (usually truncating mutations), possible striatal iron accumulation	Initial response, MF and LID rapidly occur	Widespread neuronal and glial lipofuscin accumulation without α-synuclein pathology
*RAB39B* (XLR)	Altered α-synuclein homeostasis, abnormal synapse–endosome–Golgi trafficking and recycling	From classical PD to complex syndromes with parkinsonism, dystonia, ID, globi pallidi iron deposition	Good, with frequent MF and LID	SN and cortical LBD, SN tau-positive neurofibrillary tangles
*SYNJ1* (AR)	LOF causes synaptic autophagy and transmission defects	From EOPD to atypical parkinsonism with oculomotion problems, dysarthria/anarthria, dysphagia, epilepsy, dystonia, pyramidal signs, and cognitive decline	Variable, with different adverse effects including (LID, dystonia, postural hypotension)	SN neuronal loss without LBD, neurofibrillary degeneration, tau protein staining in cell bodies and axonal hillocks
*DNAJC6* (AR)	LOF disrupts synaptic vesicle endocytosis and exacerbates α-synuclein overexpression	EOP, rapid progression with walking loss in 10 years, dystonia, pyramidal signs, postural tremor, dysarthria/anarthria, epilepsy, ID, and psychosis	Variable, frequent LID	NA
*GBA1* (AD risk factor)	Proposed disease mechanisms include ER stress, ER-Golgi transport impairment, lysosomal dysfunction with impaired α-synuclein clearance (vicious cycle), autophagy dysfunction, lipid homeostasis disruption, mitochondrial dysfunction, neuroinflammation.	Classical PD with earlier onset, faster progression, shorter survival, severe dysautonomia, depression, anxiety, and cognitive decline	Good response, MF and LID appear earlier and more frequently	SN and cortical LBD

## Data Availability

Not applicable.

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
