# Peer review of "Genetic Evidence for Endolysosomal Dysfunction in Parkinson’s Disease: A Critical Overview"

_ijms, 2023, doi:10.3390/ijms24076338_

Round 1

Reviewer 1 Report

Yahya et al have presented a detailed literature review of studies published on genetic evidence for endolysosomal defects in Parkinson’s disease (PD). The review is comprehensive and covers a number of genes associated with PD. There is no problem with this review, except it's not a critical overview. The manuscript is a summary of hundreds of published studies, with no insight/critique from the authors on what has been published. Di Fonzo and Monfrini have published in the field of PD. Therefore, I understand their expertise in the field. However, such reviews on endolysosome dysfunction, related genes and PD already exist, such as Navarro-Romero 2020, Abe 2021, and Vidyadhara 2019. I do not see anything new being contributed to the field of PD apart from summarising all the existing research papers. The authors can include the following to improve the review significantly.

Every section on genes/proteins includes a) how the gene mutations are associated with different forms of PD, b) what are associated symptoms with this particular form of PD, and c) what the protein does in health and disease. The authors can include the following more specific sections for every gene/protein to provide a more complete picture and contribute more to the PD research field.

1.     What models have been used to study a particular gene/protein

2.     What are the pros and cons that have come out of the models which has helped/not helped understanding the human disease

3.     Since the authors have expertise in PD, then why not present their own critique of the research they have cited? Providing critique will improve the quality of the review from just a summary to a more informative article that might give rise to new thinking and ideas to confront the disease pathologies.

4.     The authors have focussed on the genetic aspect of the endolysosomal system in PD. Only 10-15 % of PD cases are genetic. How about the idiopathic cases? Do they manifest endolysosomal anomalies? If so, how to explain those anomalies in the absence of mutations in genes encoding endolysosomal proteins?

5.     The figure the authors have presented has similarities with other figures published elsewhere. In fact, more detailed figures entailing the endolysosomal concept in PD have been published before. The authors need to include additional information to make the figure significantly different and informative, eventually adding new information to the PD field.

Author Response

Reply to Reviewer 1:

R: Yahya et al have presented a detailed literature review of studies published on genetic evidence for endolysosomal defects in Parkinson’s disease (PD). The review is comprehensive and covers a number of genes associated with PD. There is no problem with this review, except it's not a critical overview. The manuscript is a summary of hundreds of published studies, with no insight/critique from the authors on what has been published. Di Fonzo and Monfrini have published in the field of PD. Therefore, I understand their expertise in the field. However, such reviews on endolysosome dysfunction, related genes and PD already exist, such as Navarro-Romero 2020, Abe 2021, and Vidyadhara 2019. I do not see anything new being contributed to the field of PD apart from summarizing all the existing research papers. The authors can include the following to improve the review significantly.

A: We thank the reviewer for the insight and trust in our expertise. We agree and we modified the paper introducing a more mature view on the reported literature.

R: Every section on genes/proteins includes a) how the gene mutations are associated with different forms of PD, b) what are associated symptoms with this particular form of PD, and c) what the protein does in health and disease. The authors can include the following more specific sections for every gene/protein to provide a more complete picture and contribute more to the PD research field.

  1. What models have been used to study a particular gene/protein
  2. What are the pros and cons that have come out of the models which has helped/not helped understanding the human disease

A: We thank the reviewer for the suggestions and we agree that a complete lack of reference to in vitro and in vivo models represented a weakness of our work. Therefore, we added sections and relevant citations for each gene briefly reporting the models suggesting the respective protein function, possible implications in PD, and the strengths and limitations of the models used. A dedicated paragraph on the role of models to study human disease is now present in the conclusive paragraph.

R: 3.     Since the authors have expertise in PD, then why not present their own critique of the research they have cited? Providing critique will improve the quality of the review from just a summary to a more informative article that might give rise to new thinking and ideas to confront the disease pathologies.

A: We fully agree with the reviewer, and we are grateful for the advice. We added throughout the text several lines of our “expert opinions” on the cited literature.

R:  4.     The authors have focused on the genetic aspect of the endolysosomal system in PD. Only 10-15 % of PD cases are genetic. How about the idiopathic cases? Do they manifest endolysosomal anomalies? If so, how to explain those anomalies in the absence of mutations in genes encoding endolysosomal proteins?

A: In line with this suggestion, more sentences on idiopathic cases have been added to the text. The paragraph “Polygenic lysosomal burden" tries to address the issue of a genetic burden affecting lysosomal genes also in apparently “idiopathic” PD forms.

R: 5.     The figure the authors have presented has similarities with other figures published elsewhere. In fact, more detailed figures entailing the endolysosomal concept in PD have been published before. The authors need to include additional information to make the figure significantly different and informative, eventually adding new information to the PD field.

A: The figure has been modified and supplemented with additional information.

Reviewer 2 Report

This is an important review of pathophysiology of PD focused in the genetic endolysosomal pathway origen.

Overall being an apparently exhaustive review, it is noticeable that the authors did not follow the PRYSMA methodology to protocolize and stratify the search for information. In my opinion it would give more consistency to the review and would justify the uneven distribution of references depending on the type of gene involved.

The title of table 1:  Monogenic causes and GBA1-PD? You probably mean "Monogenic causes related to PD"  Not mentioned in the text.

The part of the discussion sections 4 and 5, which are the ones that may arouse the greatest interest in reading this review, are not very well developed. You should discuss with the most recent publications on this section. Bandres-Ciga S, et al ; International Parkinson Disease Genomics Consortium. Large-scale pathway specific polygenic risk and transcriptomic community network analysis identifies novel functional pathways in Parkinson disease. Acta Neuropathol. 2020 Sep;140(3):341-358. doi: 10.1007/s00401-020-02181-3.

Liu H, et al; International Parkinson's Disease Genomics Consortium; Comprehensive Unbiased Risk Factor Assessment for Genetics and Environment in Parkinson's Disease Consortium; Nalls M, Singleton A, Gasser T, Bandres-Ciga S. Polygenic Resilience Modulates the Penetrance of Parkinson Disease Genetic Risk Factors. Ann Neurol. 2022 Aug;92(2):270-278. doi: 10.1002/ana.26416

Foo JN, et al. Identification of Risk Loci for Parkinson Disease in Asians and Comparison of Risk Between Asians and Europeans: A Genome-Wide Association Study. JAMA Neurol. 2020 Jun 1;77(6):746-754. doi: 10.1001/jamaneurol.2020.0428.

Author Response

Reply to Reviewer 2:

R: This is an important review of pathophysiology of PD focused on the genetic endolysosomal pathway origin. Overall being an apparently exhaustive review, it is noticeable that the authors did not follow the PRYSMA methodology to protocolize and stratify the search for information. In my opinion it would give more consistency to the review and would justify the uneven distribution of references depending on the type of gene involved.

A: We thank the reviewer for the kind comment. The reason why the PRYSMA methodology was not used was that we aimed to present here a critical overview on the subject, rather than a systematic review. Nonetheless, we agree with the reviewer that the uneven distribution of references was an issue that needed to be addressed. We added relevant citations in the part where the review was most lacking.

R: The title of table 1:  Monogenic causes and GBA1-PD? You probably mean "Monogenic causes related to PD"  Not mentioned in the text.

A: We thank the reviewer for this observation. We wrote “Monogenic causes and GBA1-PD” because GBA1-PD is not unanimously deemed as a form of monogenic PD and we wanted to highlight this concept. We agree that it is not very clear and modified the text accordingly (i.e., “Monogenic causes of PD (including GBA1-PD)”).

  1. The part of the discussion sections 4 and 5, which are the ones that may arouse the greatest interest in reading this review, are not very well developed. You should discuss with the most recent publications on this section.

Bandres-Ciga S, et al ; International Parkinson Disease Genomics Consortium. Large-scale pathway specific polygenic risk and transcriptomic community network analysis identifies novel functional pathways in Parkinson disease. Acta Neuropathol. 2020 Sep;140(3):341-358. doi: 10.1007/s00401-020-02181-3 Add to Citavi project by DOI Add to Citavi project by DOI Add to Citavi project by DOI.

Liu H, et al; International Parkinson's Disease Genomics Consortium; Comprehensive Unbiased Risk Factor Assessment for Genetics and Environment in Parkinson's Disease Consortium; Nalls M, Singleton A, Gasser T, Bandres-Ciga S. Polygenic Resilience Modulates the Penetrance of Parkinson Disease Genetic Risk Factors. Ann Neurol. 2022 Aug;92(2):270-278. doi: 10.1002/ana.26416 Add to Citavi project by DOI Add to Citavi project by DOI Add to Citavi project by DOI

Foo JN, et al. Identification of Risk Loci for Parkinson Disease in Asians and Comparison of Risk Between Asians and Europeans: A Genome-Wide Association Study. JAMA Neurol. 2020 Jun 1;77(6):746-754. doi: 10.1001/jamaneurol.2020.0428 Add to Citavi project by DOI Add to Citavi project by DOI Add to Citavi project by DOI.

A: We thank the reviewer for these suggestions. These important papers were inadvertently omitted from our reference list. They are now rightfully included in our critical literature review.

Reviewer 3 Report

In this review, Yahia et al. describe the evidence that links Parkinson’s disease with endolysosomal network dysfunction.

Overall, it is a well written and conceptually clear review with a clinical viewpoint.

I only have some minor comments that may help improve the manuscript:

-        Lines 9-11: “Virtually any-membrane bound cytoplasmic organelle… associated with PD pathogenesis”.

I suggest the authors rephrasing this sentence. This sentence is not easy to understand as it is. It is not the organelle itself but its function that is modified in PD. I would rather indicate that PD pathogenesis has been related to many dysfunctional cellular mechanisms such as oxidative stress, mitochondrial dysfunction… However, most monogenic forms of PD… dysfunctional endolysosomal system…

-        I suggest the authors extending a little bit the description of what they are going to discuss in the review in the last paragraph of the introduction (lines 52-54).

I think that something similar to lines 11-17 of the abstract would guide the reader better through the review. This way, for example, when the authors mention for the first time glucocerebrosidase in the LRRK2 section (lines 90-92), the reader knows that glucocerebrosidase mutations are a risk factor for PD because he/she has read it before in the main text.

-        Section 2: monogenic causes of PD associated with endolysosomal dysfunction:

I think that the review would be more enriching if the authors made a small introduction section before starting with the description of the genetic evidence linking endolysosomal dysfunction with PD.

What I mean is that the authors could in a few lines explain the overall monogenic causes of PD (independent of their pathogenetic mechanisms) and then indicate that in this section they are going to describe specifically the ones related to endolysosomal dysfunction.

-        Line 98: it is a little confusing when the authors use the abbreviation AD PD meaning for autosomic dominant PD due to its similarity with alzheimer’s disease. I suggest the authors avoiding using it for clarity.

-        The understanding of the review text by the reader would be facilitated if the authors provided a list of abbreviations.

The authors use many abbreviations that are not easily found through the text. I think that a list of abbreviations would be helpful.

-        The review is based on a extensive list of bibliographical references. However, there are whole paragraphs that describe the clinical signs and symptoms of PD depending on the mutated gene that contain no references. I don’t know if this is because it is based on their clinical experience and not in any written document. In any case, this should be somehow clarified through the text.

-        At the end of section 2, similar to the brief introduction, in my opinion, the authors should include a small paragraph with the conclusions of the section.

-        The same for section 3: I suggest the authors including a small introduction with PD genetic risk factors (related with any mechanisms) and then focus on their relationship with the endolysosomal system. At the end of the section, I also suggest the authors including a small conclusion section. This way, the review provides a more “critical overview”, as stated in the title.

-        Section 4 the same: in my opinion it needs a couple of lines stating the generalities of what it is going to be described in the section. At the end of the section, a small integration of ideas with discussion and conclusions would help.

-        The figure is really nice and helpful to complement the information on the text. Therefore, it should be referred to somewhere in the text.

Author Response

Reply to Reviewer 3:

R: In this review, Yahia et al. describe the evidence that links Parkinson’s disease with endolysosomal network dysfunction. Overall, it is a well written and conceptually clear review with a clinical viewpoint.

A: We thank the reviewer for the appreciation of our work.

R: I only have some minor comments that may help improve the manuscript:

-        Lines 9-11: “Virtually any-membrane bound cytoplasmic organelle… associated with PD pathogenesis”.

I suggest the authors rephrasing this sentence. This sentence is not easy to understand as it is. It is not the organelle itself but its function that is modified in PD. I would rather indicate that PD pathogenesis has been related to many dysfunctional cellular mechanisms such as oxidative stress, mitochondrial dysfunction… However, most monogenic forms of PD… dysfunctional endolysosomal system…

A: We agree with the reviewer that this sentence is imprecise and unclear. We modified the manuscript accordingly.

R: I suggest the authors extending a little bit the description of what they are going to discuss in the review in the last paragraph of the introduction (lines 52-54). I think that something similar to lines 11-17 of the abstract would guide the reader better through the review. This way, for example, when the authors mention for the first time glucocerebrosidase in the LRRK2 section (lines 90-92), the reader knows that glucocerebrosidase mutations are a risk factor for PD because he/she has read it before in the main text.

A: We agree. We added a general overview clarifying the scope of the review in the introduction to help guide the readers throughout the text.

R: Section 2: monogenic causes of PD associated with endolysosomal dysfunction: I think that the review would be more enriching if the authors made a small introduction section before starting with the description of the genetic evidence linking endolysosomal dysfunction with PD. What I mean is that the authors could in a few lines explain the overall monogenic causes of PD (independent of their pathogenetic mechanisms) and then indicate that in this section they are going to describe specifically the ones related to endolysosomal dysfunction.

A: As suggested by the reviewer we added a brief introduction on PD Mendelian genetics in the introduction and monogenic forms sections.

R:  Line 98: it is a little confusing when the authors use the abbreviation AD PD meaning for autosomal dominant PD due to its similarity with alzheimer’s disease. I suggest the authors avoiding using it for clarity.

A: We removed the AD abbreviation.

R: The understanding of the review text by the reader would be facilitated if the authors provided a list of abbreviations. The authors use many abbreviations that are not easily found through the text. I think that a list of abbreviations would be helpful.

A: As requested we provided an abbreviation list.

R: The review is based on a extensive list of bibliographical references. However, there are whole paragraphs that describe the clinical signs and symptoms of PD depending on the mutated gene that contain no references. I don’t know if this is because it is based on their clinical experience and not in any written document. In any case, this should be somehow clarified through the text.

A: We thank the reviewer. It is indeed partly due to our clinical expertise in the field of monogenic forms of PD. Moreover, we condensed the references at the end of the clinical paragraph, especially for rarer genes for which the clinical phenotype is described only in a handful of case reports. Anyway, to address this issue we added several citations to the clinical descriptions. In this way, we agree, it is more scientifically rigorous.

R: At the end of section 2, similar to the brief introduction, in my opinion, the authors should include a small paragraph with the conclusions of the section. The same for section 3: I suggest the authors including a small introduction with PD genetic risk factors (related with any mechanisms) and then focus on their relationship with the endolysosomal system. At the end of the section, I also suggest the authors including a small conclusion section. This way, the review provides a more “critical overview”, as stated in the title. Section 4 the same: in my opinion it needs a couple of lines stating the generalities of what it is going to be described in the section. At the end of the section, a small integration of ideas with discussion and conclusions would help.

A: As suggested by the reviewer we added introductory and concluding paragraphs to section 2. We merged paragraphs 3 and 4 and we added an introduction and conclusion also to this new merged paragraph.

R: The figure is really nice and helpful to complement the information on the text. Therefore, it should be referred to somewhere in the text.

A: We thank the reviewer for appreciating our scientific illustration. We added several references to the figure in the text.

Reviewer 4 Report

This paper is nicely constructed review of linkages between genetic associations and complex cellular mechanisms in PD. The paper is well-written, with an excellent figure, and the references are extensive and appropriate. Publication of this fine paper recommended, with a few minimal amendments as noted below.

A few minor points.

Lines 60-61 – Suggest change to '…. years, myriad studies have confirmed.. …..'

Table 1 - Suggest changing the flow under phenotype by adding spaces between each separate phenotype, thereby visually clarifying which phenotypes relate to which gene inheritance.

Line 1297 - delete 335. No reference in the manuscript corresponds.

Author Response

Reply to Reviewer 4:

R: This paper is a nicely constructed review of linkages between genetic associations and complex cellular mechanisms in PD. The paper is well-written, with an excellent figure, and the references are extensive and appropriate. Publication of this fine paper recommended, with a few minimal amendments as noted below.

A: We are deeply grateful to the reviewer for appreciating our work.

R: A few minor points.

Lines 60-61 – Suggest change to '…. years, myriad studies have confirmed.. …..'

Table 1 - Suggest changing the flow under phenotype by adding spaces between each separate phenotype, thereby visually clarifying which phenotypes relate to which gene inheritance.

Line 1297 - delete 335. No reference in the manuscript corresponds.

A: We thank the reviewer for these suggestions. We addressed all of them in the revised version.

Round 2

Reviewer 1 Report

Nothing significant, the authors have addressed the comments.

Author Response

Reply to Reviewer 1

R: Nothing significant, the authors have addressed the comments.

A: We thank the reviewer for appreciating our reply.